# DanioCTC: Analysis of Circulating Tumor Cells from Metastatic Breast Cancer Patients in Zebrafish Xenografts

**DOI:** 10.3390/cancers15225411

**Published:** 2023-11-14

**Authors:** Florian Reinhardt, Luisa Coen, Mahdi Rivandi, André Franken, Eunike Sawitning Ayu Setyono, Tobias Lindenberg, Jens Eberhardt, Tanja Fehm, Dieter Niederacher, Franziska Knopf, Hans Neubauer

**Affiliations:** 1Department of Obstetrics and Gynecology, Heinrich Heine University of Duesseldorf, 40225 Duesseldorf, Germany; 2Center for Integrated Oncology (CIO Aachen, Bonn, Cologne, Duesseldorf), 53127 Bonn, Germany; 3Center for Regenerative Therapies TU Dresden (CRTD), Center for Molecular and Cellular Bioengineering (CMCB), TU Dresden, 01307 Dresden, Germany; 4Center for Healthy Aging, Faculty of Medicine Carl Gustav Carus, TU Dresden, 01307 Dresden, Germany; 5Anatomical Institute, Neuroanatomy, Medical Faculty, University of Bonn, 53115 Bonn, Germany; 6ALS Automated Lab Solutions GmbH, 07745 Jena, Germany; 7Life Science Center, Merowingerplatz 1 A, 40225 Düsseldorf, Germany

**Keywords:** breast cancer, circulating tumor cells, zebrafish, in vivo model, metastasis

## Abstract

**Simple Summary:**

DanioCTC is an approach that tackles a significant hurdle in cancer research: the scarcity of circulating tumor cells (CTCs) in animal models. These CTCs are crucial in understanding how cancer spreads, especially in metastatic breast cancer. To overcome the challenge of their xenotransplantation, the authors combined several advanced technologies to enrich and isolate CTCs from patients. These isolated CTCs were then injected into zebrafish embryos, allowing the researchers to study their dissemination pattern in vivo in a pilot study. Experiments showed that DanioCTC successfully enabled the xenografting of CTCs from patients. CTCs were primarily localized in the head and trunk regions of the zebrafish embryos. Overall, DanioCTC is a novel workflow to inject patient-derived CTCs into zebrafish, paving the way for a better understanding of the biology of metastatic breast cancer and the development of targeted interventions.

**Abstract:**

Circulating tumor cells (CTCs) serve as crucial metastatic precursor cells, but their study in animal models has been hindered by their low numbers. To address this challenge, we present DanioCTC, an innovative xenograft workflow that overcomes the scarcity of patient-derived CTCs in animal models. By combining diagnostic leukapheresis (DLA), the Parsortix microfluidic system, flow cytometry, and the CellCelector setup, DanioCTC effectively enriches and isolates CTCs from metastatic breast cancer (MBC) patients for injection into zebrafish embryos. Validation experiments confirmed that MDA-MB-231 cells, transplanted following the standard protocol, localized frequently in the head and blood-forming regions of the zebrafish host. Notably, when MDA-MB-231 cells spiked (i.e., supplemented) into DLA aliquots were processed using DanioCTC, the cell dissemination patterns remained consistent. Successful xenografting of CTCs from a MBC patient revealed their primary localization in the head and trunk regions of zebrafish embryos. DanioCTC represents a major step forward in the endeavors to study the dissemination of individual and rare patient-derived CTCs, thereby enhancing our understanding of metastatic breast cancer biology and facilitating the development of targeted interventions in MBC. Summary statement: DanioCTC is a novel workflow to inject patient-derived CTCs into zebrafish, enabling studies of the capacity of these rare tumor cells to induce metastases.

## 1. Introduction

Breast cancer metastasis is a complex process involving tumor-cell migration, intravasation, survival in the bloodstream, adhesion, extravasation, and colonization at secondary sites [1,2]. Metastatic breast cancer (MBC) is the major cause of breast cancer morbidity and mortality, and identifying the mechanisms underlying the metastatic process and developing clinical strategies to detect and treat MBC patients are crucial [3].

Circulating tumor cells (CTCs) are shed into the blood by tumor tissue and are considered precursor cells for metastasis formation [4]. Despite being extremely rare, CTC numbers correlate with poor survival outcomes in metastatic and non-metastatic cancers, making their isolation and analysis important for therapy prediction and for understanding their function [5,6,7,8,9]. CTC analysis also enables the monitoring of the treatment response [10].

One technology to increase the number of informative cancer patient samples containing high numbers of analyzable CTCs is diagnostic leukapheresis (DLA), a density-based blood separation that facilitates the safe collection of large CTC numbers from liters of patient blood [11]. DLA enables CTC-positivity rates of more than 90% and a 30-fold increase in CTC numbers compared to normal blood volumes [7,12,13]. Moreover, DLA-isolated CTCs are viable and can be used in in vitro assays or for the development of preclinical animal models [14], both of which are essential for investigating the metastatic biology of CTCs and for initiating novel effective treatment strategies.

The limitations of the current mouse xenotransplantation models for the study of early tumor dissemination have prompted the need for alternative animal models. Zebrafish have emerged as a valuable tool for investigating human diseases, including cancer metastasis, owing to their genetic similarity to humans, ease of maintenance, and high fecundity [15,16,17]. Zebrafish embryos and larvae are transparent, enabling non-invasive experimental procedures such as high-resolution in vivo microscopy, which provides insights into cancer cell distribution, invasion, and proliferation [16,18]. Although zebrafish xenotransplantation models have been developed to study various malignancies [19], the experimental requirements of microinjection for xenotransplantation currently rely on high tumor cell numbers and/or (transient) in vitro cultivation, which limit the use of zebrafish (as other animal models) in the study of CTCs [20,21]. To overcome this limitation and to expand the repertoire of xenotransplantation approaches in zebrafish, we have developed a novel workflow for injecting a few isolated CTCs from MBC patients into zebrafish embryos. This approach provides an important model to investigate the CTCs’ metastatic potential in zebrafish, addressing one of the requirements to develop personalized therapy approaches and facilitating the discovery of new therapeutic strategies for metastatic cancer treatment.

## 2. Methods

### 2.1. Cell Line and Culture Conditions

MDA-MB-231 cells were obtained from the American Type Culture Collection (Manassas, VA, USA); EGFP-labeled MDA-MB-231 cells were kindly provided by Manja Wobus/Martin Bornhäuser (University Hospital Carl Gustav Carus, TU Dresden, Dresden, Germany). MDA-MB-231 cells were cultured in RPMI 1640 medium, containing 10% fetal bovine serum, 25 mM HEPES (Gibco, Grand Island, NY, USA) and 100 units/mL penicillin-streptomycin in a humidified incubator at 37 °C with 5% CO_2_; EGFP-labeled MDA-MB-231 cells were cultured in low glucose DMEM (1 g/mL D-Glucose, Gibco) with 10% fetal bovine serum (Gibco) in a humidified incubator at 37 °C with 5% CO_2_. Cultured cells were harvested at a confluence of approximately 80% for staining and spike-in experiments (i.e., experiments in which tumor-cell-free samples were supplemented with tumor cells).

### 2.2. Zebrafish and Housing Conditions

The zebrafish (*Danio rerio*) strains were maintained according to national law and under standardized conditions as previously described [22,23] (licenses DD25-5131/450/4 and 25-5131/564/2). Lines used in this study were Tg(*kdrl*:EGFP) [24] kindly provided by Dr. Bernhard Fuss/Prof. Dr. Didier Stainier and Tg(*osx*:mCherry) [Tg(*OlSp7*:mCherry)^zf131^] for initial characterization of MDA-MB-231 cancer cell distribution in zebrafish embryos [25].

### 2.3. Standard Injection of Cell Line Cells into Zebrafish Embryos

EGFP labeled MDA-MB-231 cells were trypsinized using trypsin–EDTA (0.25%), neutralized using complete medium, centrifuged at 1200 rpm for 10 min, resuspended in PBS, and additionally stained with SP-DiOC18(3) (D7778, Invitrogen, Waltham, MA, USA) according to the manufacturer’s recommendations, in order to also stain isolated cells that might have lost their GFP fluorescence (clonal vector loss) during cell division. The stained cell suspension was centrifuged for 5 min at 1200 rpm and the cell pellet suspended in transplantation buffer [0.9× PBS, 1.5 mM EDTA (Roth, Karlsruhe, Germany), 1% penicillin-streptomycin (Roth)] at 100–150 cells/nl. Glass needles for injection were pulled from glass capillaries with filament (TW100F-3, WPI, Sarasota, FL, USA) with a P-97 Flaming/Brown micropipette puller (Sutter Instrument Co., Novato, CA, USA). Zebrafish embryos (2 days post fertilization, dpf) were manually dechorionated, anesthetized using 0.02% tricaine and transferred into a petri dish which was covered with a thin layer of 1.5% low melting agarose in E3 (50 mM sodium chloride, 0.17 mM potassium chloride, 0.33 mM calcium chloride, 0.33 mM magnesium sulfate), in order to avoid adhesion of the embryos to the dish. The labeled MDA-MB-231 cell suspension was loaded into a pulled glass capillary and 1–2 nL were micro-injected into the blood circulation of Tg(*osx*:mCherry) zebrafish embryos via the duct of Cuvier (DoC) using a pneumatic PicoPump (SYS-PV820, WPI, Sarasota, FL, USA). Engrafted embryos were maintained in a new petri dish at 33 °C [26]. Based on the fluorescence spread of the injected embryos post injection, embryos with tumor cells in the blood circulation were selected for maintenance for up to three days post injection (dpi). Anesthetized zebrafish embryos were imaged daily starting on the day of injection with an Olympus MVX10 microscope to inspect survival, cardiac edema, homing of tumor cells and extravasation events. The imaging time was kept to a minimum to ensure minimal exposure to anesthesia. No effect on developing larvae was observed during the experimental period, except for one larva which did not survive up to 3 dpi and which was therefore excluded from the analysis (n = 6 throughout the experiment). Data of 1 dpi and 3 dpi are shown. For analysis, the larval body was divided into three regions: head, trunk, and tail (Appendix A). Quantification of absolute numbers of cell dissemination was performed using the cell counter feature in Fiji.

### 2.4. Workflow for Injection of CTCs into Zebrafish Embryos

#### 2.4.1. Diagnostic Leukapheresis

Diagnostic leukapheresis (DLA) was performed at the Department of Transplantation Diagnostics and Cell Therapeutics, Duesseldorf, Germany, as previously described [11,13]. Aliquots of the DLA sample were frozen and stored for future use. Cryopreservation of DLA samples was described by our group before [14].

#### 2.4.2. Enrichment of Viable CTCs or Spiked Cell Line Cells from DLA Samples with the Parsortix System

Cryopreserved DLA samples were rapidly thawed in a 37 °C water bath and filtered through a fine sieve (mesh size 100 µm). The filtered DLA sample was diluted 1:20 in PBS. For spike-in experiments, MDA-MB-231 cell line cells were added to diluted negative control DLA samples. Therefore, cultured MDA-MB-231 cells were trypsinized using trypsin–EDTA (0.05%), neutralized using complete medium, centrifuged at 1200 rpm for 5 min and resuspended in PBS. Subsequently, 5000 MDA-MB-231 cell line cells were transferred into diluted negative control DLA samples. Viable CTCs of a MBC DLA sample or spiked cell line cells of negative control DLA samples were enriched with the FDA-approved Parsortix system (Angle plc, Guildford, UK). The Parsortix system was used according to the manufacturer’s instructions. Following the protocol, filtration cassettes with 6.5 µm gaps were used and 100 mbar above ambient pressure were applied [14,27].

#### 2.4.3. Isolation of Cell Line Cells and CTCs by Flow Cytometry and Staining for Cell Tracking

Parsortix pre-enriched MDA-MB-231 cell line cells or CTCs were further isolated by flow cytometry. Therefore, enriched MDA-MB-231 cells or CTCs were stained in parallel with EpCAM (1:50, VU1D9, Stemcell^TM^ Technologies, Vancouver, BC, Canada, AF488), MUC1 (1:25, CD227, eBioscience^TM^, Santa Clara, CA, USA, AF488), Her2 (1:50, CellSearch CXC kit, Menarini Group, Florence, Italy, AF488) and CD45 (1:25, 3S-Z5, Santa-Cruz Biotechnology, Dallas, TX, USA, AF647) at 37 °C for 60 min for identification by flow cytometry. EpCAM-mediated enrichment of CTCs for xenotransplantation has previously been used [28,29]. Cell suspensions were washed twice with PBS at 1500 rpm for 4 min and were resuspended in PBS. The protocol used for flow cytometry consisted of a discrimination of FITCpos/neg and CD45pos/neg events. FITCpos/CD45neg cells were sorted into a 1.5 mL tube. Subsequently, cells were additionally stained with 10 μM CellTracker Red CMPTX (C34552, Thermo Fisher Scientific, Waltham, MA, USA) at 37 °C for 30 min for cell tracking in zebrafish larvae. Cell suspensions were washed twice with PBS at 1500 rpm for 4 min and resuspended in transplantation buffer.

#### 2.4.4. Detection and Picking of Single CTCs and Cell Line Cells Using the CellCelector

Detection and isolation of single stained CTCs and MDA-MB-231 cell line cells were performed with the CellCelector (ALS GmbH, Jena, Germany) [30], which is a semi-automated micromanipulator consisting of an inverted fluorescent microscope (CKX41, Olympus, Tokyo, Japan) with a CCD camera system (XM10-IR, Olympus, Tokyo, Japan) and a robotic arm equipped with a vertical glass capillary. A 20 µm capillary (CC0048, ALS GmbH, Jena, Germany) with a broken tip was used for cell isolation as well as injection. Cell suspensions were transferred onto ALS MagnetPick glass slides (Sartorius, Göttingen, Germany, CC0059), placed on the automatic stage of the CellCelector microscope and were allowed to settle. Samples were manually scanned with 20× and 40× magnification using the following channels: brightfield (BF, cell morphology), FITC (EpCAM), TRITC (CellTracker) and Cy5 (CD45). The following exposure time was used: 200 ms (FITC and Cy5). Fifty single cells were picked by the capillary that met the criteria: round, intact morphology with a cell diameter of 4–40 µm in BF, FITC positive/TRITC positive/Cy5 negative. The cells were allowed to settle in the capillary tip for 30 min before injection into the DOC of a 2 dpf Tg(*kdrl*:EGFP) zebrafish embryo, in order to minimize the injection volume. The procedure was repeated for several embryos.

#### 2.4.5. Injection of Stained CTCs and Cell Line Cells into Zebrafish Embryos by Combining the CellCelector with a Stereomicroscope

Transgenic *Tg*(*kdrl:EGFP*) zebrafish embryos (2 dpf) were manually dechorionated and anesthetized using 0.02% tricaine and transferred to a petri dish covered with 1.5% low melting agarose in E3. The petri dish with the transgenic *Tg*(*kdrl:EGFP*) zebrafish embryos was positioned in the deposit area of the CellCelector. An additional Nexius Zoom EVO stereomicroscope with a flexible arm was installed to visualize the embryos in the deposit area of the CellCelector. Approximately 5–10 nL of the capillary volume, containing the 50 picked, settled cells in the capillary tip, were semi-automatically micro-injected with the CellCelector into the DoC of individual zebrafish. Engrafted embryos were maintained in a new petri dish filled with E3 medium at 34 °C. Based on the fluorescence spread of the injected embryos at 2 h post injection (hpi), embryos with tumor cells in the blood circulation were selected and each xenografted zebrafish embryo was transferred into a single dish filled with E3 medium for maintenance for up to 3 dpi. Zebrafish embryos were imaged at 1 dpi and 3 dpi with the fluorescent microscope of the CellCelector to inspect survival, cardiac edema, homing of tumor cells and extravasation events. The imaging time was kept to a minimum to ensure minimal exposure to anesthesia. No effect on developing larvae was observed during the experimental period.

The complete duration of the workflow—from CTC enumeration to zebrafish CTC injection—was 3–4 h.

### 2.5. Patient Samples

Patient samples were selected from the Augusta study (approved by the Ethics Committee of the Medical Faculty of the Heinrich Heine University Düsseldorf, Ref. No. 3430). Analysis of human samples was carried out in accordance with the Good Clinical Practice guidelines. All patients provided written informed consent for the use of their blood samples for CTC analysis and for translational research projects. Injected CTCs were isolated from a 57-year-old patient with a hormone-receptor positive, Her2/neu negative breast cancer with bone, bone marrow and lymph node metastases.

### 2.6. Statistics

Statistical analysis was performed using the Prism analysis program (GraphPad Inc., San Diego, CA, USA). Data are expressed as mean ± SD. Statistical analyses were performed for parametric data by ANOVA followed by post hoc Bonferroni test. For non-parametric data, statistical analyses were performed by Kruskal–Wallis test followed by post hoc Dunn’s test, in order to correct for multiple comparisons. *p* < 0.05 was considered to be statistically significant (* 0.01 < *p* < 0.05; ** 0.001 < *p* < 0.01; *** 0.0001 < *p* < 0.001).

## 3. Results

### 3.1. Development of a CTC Injection Workflow—Concept and Challenges

We set up an injection workflow for low CTC numbers by combining, adapting and fully integrating existing, well-validated workflows for each of the required technology modules: (i) xenotransplantation of cancer cells into zebrafish embryos by microinjection into the blood circulation; (ii) DLA, which allows large volumes of blood to be processed for CTC analysis; (iii) flow cytometry for the provision of well-characterized CTC preparations; and (iv) the CellCelector automated micromanipulation system—extended by an additional optical system—for the isolation of individual cells and performing their collective injection. It is worth mentioning that although injection of cells into zebrafish is an already well-established method, high cell number preparations are currently required for xenotransplantation, restricting their use to cell lines or transiently cultured primary cells. The main challenge for the establishment of the presented DanioCTC approach was to fine-control the injection process and to optimize the design of the glass capillary used for picking and injecting selected CTCs at desirable injection rates while maintaining acceptable health conditions for the embryos. In addition, live-cell labelling and fluorochrome detection for tracking the injected cells in the living zebrafish embryo were improved, as CTCs, due to the challenges involved in cultivating them, cannot be efficiently transfected, in contrast to cell lines. In addition, the direct injection of CTCs without any preculture is preferred to inspect their function during metastasis.

We followed the above concept by first using the invasive, triple-negative MDA-MB-231 cell line to establish the microinjection procedure and to test for the cells’ distribution and survival in zebrafish embryos. This was followed by the development of the entire DanioCTC workflow including the cell line’s live-cell labelling and isolation from spiked-in DLA samples for their in situ tracking. Ultimately, the workflow was adapted and used to inject and monitor patient-derived CTCs (Figure 1).

### 3.2. Adaptations That Were Required to Realize DanioCTC

#### 3.2.1. Optimizing Cell Picking and Injection

In order to determine the best-suited capillary type for injecting the CTCs into the zebrafish larvae, the commercially available capillary of the CellCelector was compared with other capillaries, which were prepared with a micropipette puller and regularly used for zebrafish injections (Figure 2). The self-prepared capillary was characterized by a long taper and a slightly bigger opening diameter (ca. 25 µm). The original CellCelector capillary, on the other hand, tapered quickly, was therefore rather short and its opening had a diameter of 20 µm. Since more cells stuck in the longer tapered tip of the self-prepared capillary, the use of this capillary type was discontinued and the CellCelector capillary was used. To improve the penetration of the CellCelector capillary into the zebrafish embryo, the tip was broken off with forceps increasing its diameter to approx. 25 µm. An angle of approximately 45° for the capillary opening proved to be the most effective for penetrating zebrafish tissue.

#### 3.2.2. Adaptation of the CellCelector Setup for Cell Injection

In order to enhance the functionality of the CellCelector, we collaborated with the manufacturers to modify the CellCelector software (Version 3.0). We programmed the joystick that allowed manual control of the robotic arm and injection process. Additionally, we created an automated injection process that enabled adaptation of the injection speed and volume. To simplify the workflow, we defined a fixed position for the zebrafish larvae and established a motion sequence for the robotic arm during injection.

#### 3.2.3. Installation of a Stereomicroscope at the CellCelector

One challenge in the injection procedure was to ensure that the entire process could be observed. As the CellCelector system uses an inverted microscope, it was not initially suitable for imaging from the top as required for injection into the DoC. To overcome this, we extended the system by installing a flexible stereomicroscope at the programmed deposition area of the CellCelector robotic arm. This allowed us to localize the DoC and position the capillary with precision. By aligning the CellCelector capillary with the DoC, we were able to achieve complete visualization of the injection process (as shown in Figure 3).

### 3.3. Standard Xenotransplantation Workflow Carried out with MDA-MB-231 Cells

In our experiments, we used the standard workflow to inject EGFP-labeled MDA-MB-231 cells into 2-day-old zebrafish embryos. The injected cells dispersed into various areas of the zebrafish embryos, including the head, trunk, and tail, which contains the caudal hematopoietic tissue (CHT), the site of blood formation at this larval stage (Figure 4). At 1 dpi, an average of 38.6 ± 14.8 cells disseminated into the head region, 30 ± 15.5 cells into the trunk region, and 123 ± 31.4 cells into the tail region. However, by 3 dpi, only 30 ± 20.5 cells were located in the head, 16.3 ± 10.2 cells in the trunk, and 46.3 ± 19.5 cells in the tail. This suggests that approximately 50% of the injected cells died and disappeared between 1 and 3 dpi, with more cells getting lost in the trunk and tail regions compared to the head region. Relative cell numbers can be found in Appendix A.

### 3.4. Injection of MDA-MB-231 Cells Spiked into a DLA Aliquot with the DanioCTC Workflow

Since the CTCs were to be isolated from DLA products obtained from MBC patients, we mimicked such a sample by spiking about 5000 MDA-MB-231 cells into a CTC-negative 2 mL DLA aliquot (negative control) in the next step and processed it according to the workflow depicted in Figure 1 and Figure 5. Per embryo [Tg(*kdrl*:EGFP)], 50 pre-labeled MDA-MB-231 cells were injected with the CellCelector, resulting in a total of 11 embryos. The zebrafish embryos were incubated and the injected cells were monitored at 1 and 3 dpi. The MDA-MB-231 cells disseminated to different areas in the head, trunk, and tail regions in varying numbers (Figure 5A–D). On average, 3.5 ± 3.2 cells disseminated into the head, 0.9 ± 1.0 cells into the trunk, and 5.9 ± 4.7 cells into the tail at 1 dpi (Figure 5D). At 3 dpi, 3.8 ± 2.0 cells were detected in the head, 0.8 ± 1.1 in the trunk, and 3.9 ± 4.4 in the tail, which is equivalent to approximately 83% of the cell numbers present at 1 dpi. The relative cell numbers are depicted in Appendix A. The MDA-MB-231 cells did not only accumulate but also partially extravasate from the blood circulation into surrounding tissue areas in the CHT and other tail regions (Figure 5C). This finding is consistent with the accepted metastatic nature of the MDA-MB-231 cell line [26,31,32].

### 3.5. Injection of CTCs Isolated from DLA Aliquots with the DanioCTC Workflow

A cryopreserved 2 mL DLA aliquot obtained from a patient with MBC was used for CTC enrichment using the Parsortix system, following our workflow (Figure 1). A total of 50 CTCs per zebrafish embryo were injected (n = 9).

After injection, CTCs disseminated to different areas in the head, trunk, and tail (Figure 6), with an average of 0.4 ± 0.7 CTCs in the head, 2.2 ± 2.2 CTCs in the trunk, and 0.4 ± 0.7 CTCs in the tail at 1 dpi. At 3 dpi, the mean number of CTCs detected in the head was 1.8 ± 1.7, in the trunk was 0.6 ± 1.0, and in the tail was 0.2 ± 0.6, which is equivalent to approximately 87% of the cell numbers present at 1 dpi. No significant differences between groups were observed. We did not observe any extravasation events at 1 dpi or 3 dpi. See Appendix A for relative cell numbers.

## 4. Discussion

Here, we introduce DanioCTC, a seamless workflow that allows for the injection of low numbers of patient-derived CTCs into zebrafish embryos. This technique enables real-time studies of the differential metastatic traits of individual CTCs in vivo, despite their scarcity in cancer patients’ blood. Unlike previous methods which rely on high cell numbers and retrograde loading of cells into glass capillaries, our approach overcomes this technical limitation by integrating the injection process, viable CTC isolation, labelling, and detection into a reproducible workflow. In general, DanioCTC provides the methodology to investigate cells with a low prevalence in zebrafish embryos, as shown here with patient-derived CTCs. The workflow allows for the selection of cells of interest down to the single-cell level and their subsequent injection. Consequently, the method paves the way for studies investigating the dissemination of cells in vivo, along with studies focusing on the molecular characterisation of cells after injection. With DanioCTC, we aim to preserve the competency of CTCs to metastasize by using directly isolated CTCs. Our workflow combines the detection of CTCs from DLA products with ex vivo labelling techniques and direct injection using the automated micromanipulation system CellCelector, which we have modified and extended. DanioCTC facilitates the investigation of CTCs during the process of metastasis by enabling the injection of low numbers of patient-derived CTCs into zebrafish embryos. This approach provides a good basis for modelling the low frequency of CTCs typically found in cancer patients’ blood. The zebrafish embryo is an ideal in vivo model for studying CTC metastasis due to its transparency, fast development, high progeny number, and lack of adaptive immunity in initial life stages. In this transparent model, the capacity of CTCs to arrest and adhere to the endothelium, which may involve cell rolling and other processes that can be observed in tumor cells and immune cells found in the circulation [33], can be assessed in a dynamic fashion, e.g., by high-resolution video microscopy. In addition, the hemodynamic forces of blood flow that impact the arrest of CTCs can be studied, e.g., by monitoring different locations in the embryo displaying distinct flow rates, and manipulating flow rates with heartbeat-modifying drugs [34]. Compared to in vitro assays, which are somewhat static in nature and may not accurately reflect the low affinity and transient interactions of CTCs with endothelia, DanioCTC will provide a more accurate way to monitor CTC transport and migration in vivo. In combination with other advantages (transparency, fast development, high progeny number, and lack of adaptive immunity in initial life stages) zebrafish represent an ideal in vivo model to study CTC metastasis.

### 4.1. MDA-MB-231 Cell and CTC Distribution in Zebrafish Larvae

We present the results of injecting MDA-MB-231 cells at both high and low cell numbers into zebrafish embryos, where they were frequently found in the head (including the CNS) and tail (CHT), which is consistent with the high incidence of brain and bone marrow metastases in triple negative breast cancer [35]. The relative cell numbers in the head and tail changed over time, indicating a higher survival rate of MDA-MB-231 cells in the head. However, further investigations are needed to determine which individual tumor cells survive, as the redistribution of cells over time suggests variability in this aspect. MDA-MB-231 cells were observed to extravasate from blood vessels into the surrounding tissue, which is in agreement with previous findings [26,31]. In contrast, CTCs from a MBC patient injected into zebrafish larvae were primarily located in the head and trunk, unlike MDA-MB-231 cells that were present in the head and tail. The relevance and reasons for this are currently unclear. Flow conditions in the blood vessels of the respective domains, as well as the size or markers of the cells, might influence the outcome [19]. In addition, heterogeneity in CTCs might be high at the DNA level [36]. However, caution is warranted as our observation is based on a single experiment and DLA aliquot. Therefore, further investigation is required to determine if there is a correlation between brain metastases and the localization of CTCs to the head in zebrafish embryos. In the future, it will be important to isolate tumor cells from different regions of injected zebrafish embryos to compare their gene expression profiles with CTCs and disseminated tumor cells (DTCs) from the patient to examine their expression profiles and clonal relationship. Furthermore, more MBC patient-derived samples should be investigated with the DanioCTC workflow.

### 4.2. Advantages and Limitations of DanioCTC

Our study, for the first time, employs a semi-automated single-cell micromanipulation system to demonstrate that CTCs from breast cancer patients can be directly injected into zebrafish embryos and monitored for several days by live imaging. While traditional mouse models have been utilized to mimic different stages of the metastatic cascade, zebrafish models offer a promising alternative. Previous research has showcased the potential of zebrafish models in investigating the biology of non-patient CTCs and CTC-clusters, indicating their utility in understanding the metastatic nature of cancer cells [20,26,31]. For example, Berens et al. examined cancer cell extravasation in the zebrafish embryo tail 96 h after injection with different cell lines [31]. Zebrafish embryos injected with the MDA-MB-231 cell line exhibited the greatest number of extravasated cancer cells per embryo, which is in line with its increased metastatic nature. In agreement with this, Asokan et al. showed that MDA-MB-231 cells were capable of invading avascular fin fold tissue while normal breast epithelial cells were not [26]. At 4 dpi, 31.8% of injected MDA-MB-231 cells had extravasated into the caudal fin fold. Martinez-Pena and colleagues revealed that CTC-clusters from MDA-MB-231 cells disseminated at a lower frequency than single MDA-MB-231 CTCs in the zebrafish, showed a reduced capacity to invade but had a higher survival and proliferation capacity in the temporal follow-up than single CTCs [20].

Despite those advancements, cell lines as ‘surrogate CTCs’ have been used in all previous studies to investigate the differential behavior of single CTCs and CTC clusters in zebrafish embryos, due to limitations in the injection process. It is important to note that subtle differences have been observed between cell line-derived surrogate CTCs and patient-derived CTCs [37,38,39] and caution should be exercised when interpreting the results.

Here, we provide a proof-of-concept study on the DanioCTC workflow that allowed us to inject isolated CTCs from a single MBC patient into zebrafish embryos, to the best of our knowledge, for the first time. This case study, along with the established setup, will allow for further experiments on isolated CTCs, accompanied by ongoing improvements in the equipment and software. Future enhancements in the procedure and setup will reduce the processing time and increase the applicability of the method in personalized medicine approaches. Our case study using DanioCTC paves the way for future investigations of MBC-patient cohort CTCs by studying and characterizing the distribution of these cells in vivo.

In summary, our study establishes an innovative workflow for the injection of MBC-derived isolated CTCs into zebrafish embryos, enabling live imaging of patient-derived CTC dissemination in xenografts over time. This approach may become a valuable tool in further studies exploring targeted medicine approaches and the in-depth exploration of CTC metastatic biology, given its capability to work with only a few dozen cells per embryo.

## Figures and Tables

**Figure 1 cancers-15-05411-f001:**
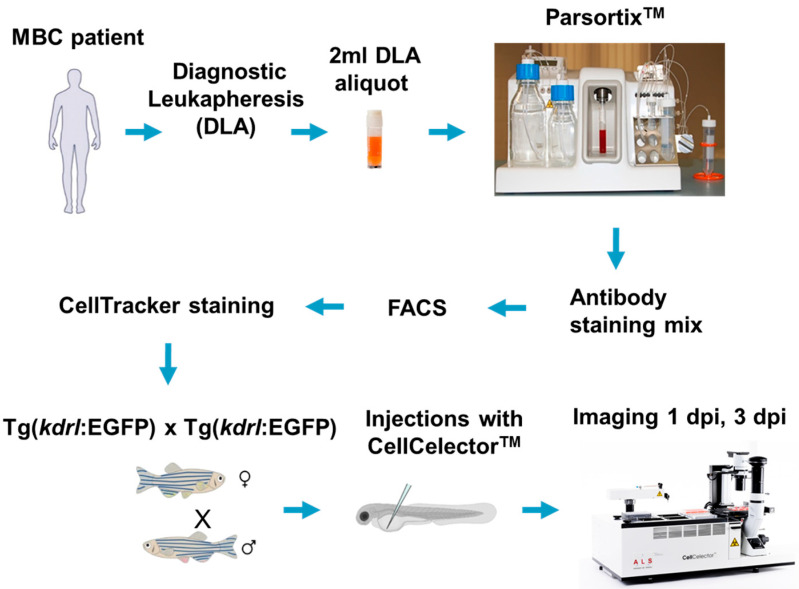
DanioCTC Workflow. A MBC patient undergoes diagnostic leukapheresis (DLA). CTCs are enriched from a DLA aliquot with the Parsortix system and isolated by FACS after staining. Cells are further stained by CellTracker Red for tracking in zebrafish embryos. Single CTCs are isolated by using the CellCelector and injected into 2 dpf old Tg(*kdrl*:EGFP) zebrafish embryos. Adapted from [27,30].

**Figure 2 cancers-15-05411-f002:**
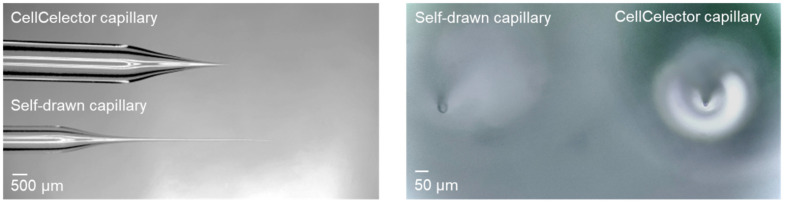
Capillaries. Self-prepared capillary for standard zebrafish embryo injections with a long taper narrowing and an opening diameter of approximately 25 µm. The CellCelector capillary tapers quickly, is rather short and has an opening diameter of 20 µm.

**Figure 3 cancers-15-05411-f003:**
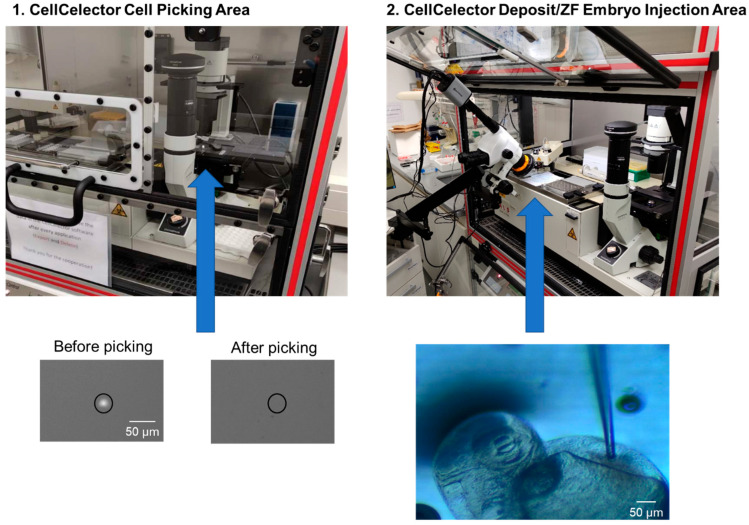
Adapted CellCelector setup. Representative images of the CellCelector picking area of picking a single CellTracker Red-stained CTC and the attached stereomicroscope required for CTC injections in the deposit area of the CellCelector. (1) CellCelector cell picking area and image of region of interest before and after cell picking. (2) CellCelector deposit/zebrafish injection area and brightfield image of embryo being injected.

**Figure 4 cancers-15-05411-f004:**
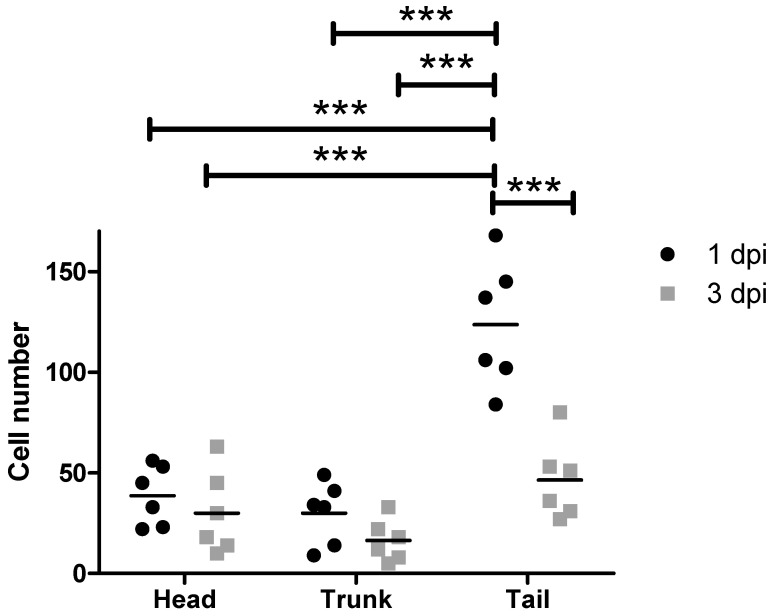
Dissemination of MDA-MB-231 cells after injection with the standard workflow into zebrafish larvae. Depicted are the absolute MDA-MB-231 cell numbers in the head, trunk and tail at 1 and 3 dpi (n = 6). dpi: days post injection. ANOVA followed by post hoc Bonferroni test, *** 0.0001 < *p* < 0.001.

**Figure 5 cancers-15-05411-f005:**
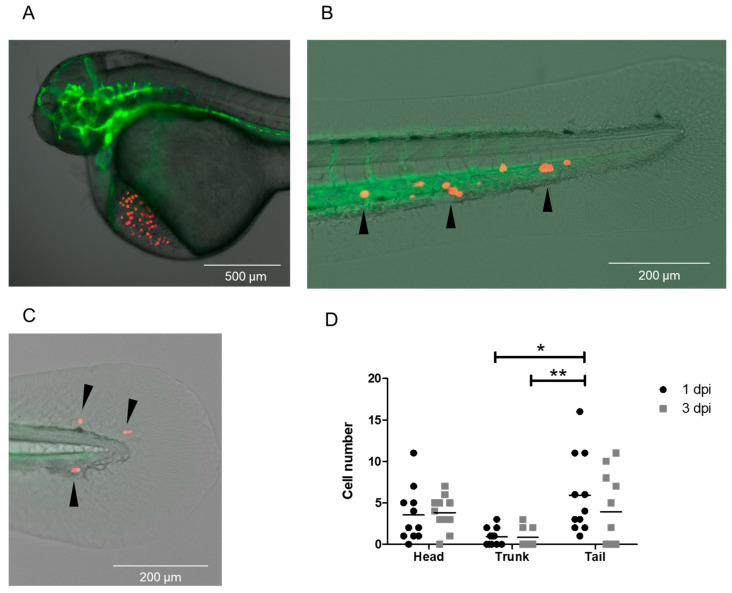
Dissemination of MDA-MB-231 cells spiked into a DLA sample after injection with the DanioCTC workflow. Cell localization was monitored at 1 and 3 dpi. (**A**) Representative image of 50 MDA-MB-231 cells injected into the DoC of a zebrafish embryo (1 hpi). (**B**) Representative image of the tail and CHT region at 3 dpi with localized MDA-MB-231 cells in the CHT (black arrowheads). (**C**) Representative image of the tail region at 3 dpi showing extravasated MDA-MB-231 cells (black arrowheads). (**D**) Absolute MDA-MB-231 cell numbers detected in the head, trunk and tail regions at 1 and 3 dpi (n = 11 embryos). Green label: endothelium; red label: MDA-MB 231 cells. dpi: days post injection. Kruskal–Wallis test followed by post hoc Dunn’s test, * 0.01 < *p* < 0.05; ** 0.001 < *p* < 0.01.

**Figure 6 cancers-15-05411-f006:**
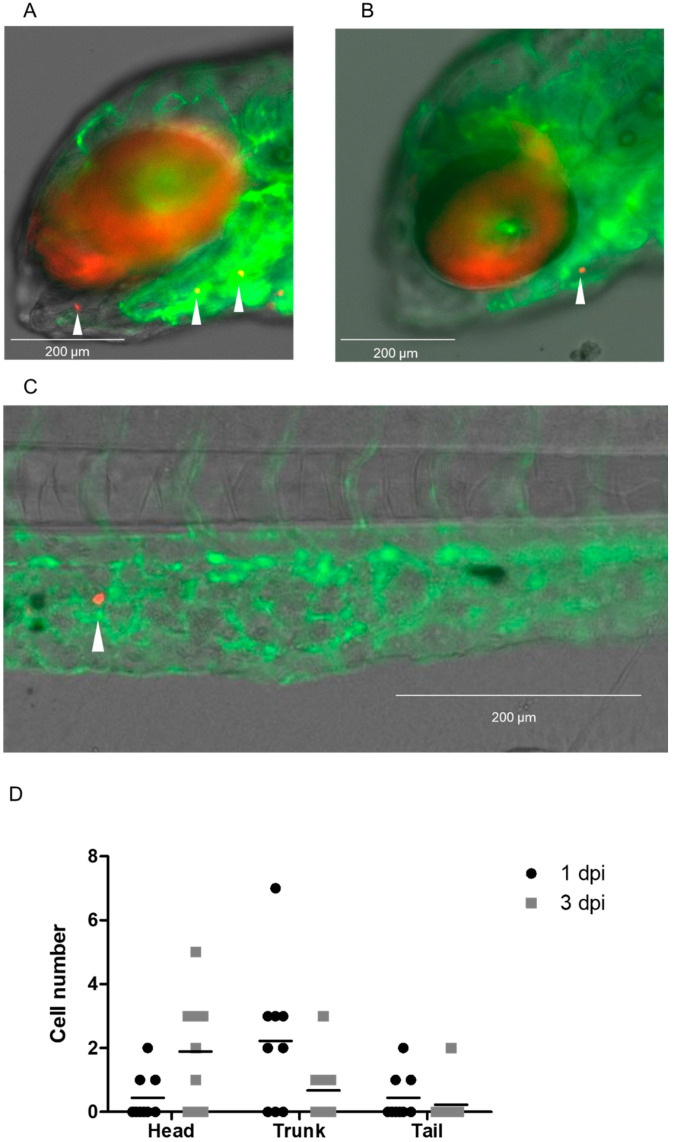
Dissemination of isolated CTCs of a MBC patient after injection with the DanioCTC workflow. Isolated CTCs, labeled in red, were monitored at 1 and 3 dpi and showed dissemination into the head, trunk and the tail. (**A**,**B**) Representative images of the head region of a Tg(*kdrl*:EGFP) positive embryo after injection of 50 CTCs (indicated by white arrowheads). (**C**) Representative image at 3 dpi showing a disseminated CTC in the CHT of an embryo (indicated by white arrowhead). (**D**) Absolute CTC numbers in the head, trunk and tail at 1 and 3 dpi (n = 9). dpi: days post injection. Kruskal–Wallis test followed by post hoc Dunn’s test. No statistically significant differences were observed between groups.

## Data Availability

The data generated in this study are available upon request from the corresponding authors.

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
