# Peer review of "DanioCTC: Analysis of Circulating Tumor Cells from Metastatic Breast Cancer Patients in Zebrafish Xenografts"

_cancers, 2023, doi:10.3390/cancers15225411_

Round 1

Reviewer 1 Report

Comments and Suggestions for Authors

This manuscript presents a proof-of-concept describing the workflow allowing to inject & track CTCs in zebrafish. The imaging system offers high quality images of detectable CTCs. The authors demonstrated the feasibility of this procedure using two sources of tumor cells (MDA-MB 231 cells and DLA from MBC patient). Even if the work is interesting, the manuscript describes preliminary data and there are limited novelties. The inoculation of fluorescent breast cancer cell line has been already published and the novelty is mainly focused on the inoculation of CTCs isolated from patient samples, unfortunately only one patient sample has been studied with no molecular characterization. Indeed, Parsortix is a pre-enrichment system for CTCs and is based on the bigger size and lower deformability of CTCs and not on the expression of biological markers. However, the authors described the CTCs isolated from patients as EpCAM positive and CD45- (there is no description of the method used). Between the pre-enrichment step and the injection of CTCs, tumor cells were then labeled with specific antibodies. What was the impact of this staining on the ability of cells to migrate in embryos (this is a key question)?

It is definitely mandatory to increase the number of  patient samples, especially due to the difference of distribution between MDA & CTC and the variability of their size and general properties.

In addition, the use of a single cell line is an issue, and various cell lines (size and origin) should be compared to better understand the effects of their characteristics in the process.

Brief data are shown and many questions arise concerning the survival of cells in the longer term, their organization (cluster formation, mobility,...). A better characterization of the process is  mandatory.

It seems that even with groups of 9 embryos the variability is significant. How to explain this massive vaiability ?

This manuscript shows a system to study CTC in zebrafish but this work offers few original information and requests complementary data.

Author Response

Reviewer #1

1) This manuscript presents a proof-of-concept describing the workflow allowing to inject & track CTCs in zebrafish. The imaging system offers high quality images of detectable CTCs. The authors demonstrated the feasibility of this procedure using two sources of tumor cells (MDA-MB 231 cells and DLA from MBC patient). Even if the work is interesting, the manuscript describes preliminary data and there are limited novelties. The inoculation of fluorescent breast cancer cell line has been already published and the novelty is mainly focused on the inoculation of CTCs isolated from patient samples, unfortunately only one patient sample has been studied with no molecular characterization.

Response: The reviewer is correct in his assessment that work has been published on the inoculation of tumor cell lines into zebrafish larvae (e.g. [1]). Indeed, we made use of the current knowledge on cell line injection setups to develop our workflow. Our work’s importance lies in the development of a combined individual cell-isolation and injection approach which has not been available to the cancer research community before. Isolating patient CTCs and injecting them in very low numbers into an in vivo animal model is not trivial but technically challenging, in particular considering the limitations of classic models (rodents). The need of injecting high amounts of cells into mammalian models precluded investigation of primary CTCs that had not been grown in culture after isolation from blood samples. Instead, current techniques require the injection of hundreds of thousands of cells, mainly due to retrograde loading of cell suspensions into capillaries. These current approaches entirely negate the low availability of CTCs and the need to select them at the single cell level. A PubMed search with the keywords [CTC AND zebrafish] results in 16 hits including two recent publications from the group of Pineiro [2, 3] which are investigating CTCs in zebrafish. Due to the mentioned limitations of the workflow cell lines had to be used in these studies. In addition to establishing and demonstrating the injection procedure for individual cells by using an anterograde loading of capillaries (cell picking), DanioCTC also describes the methodology preceding injection (i.e. CTC isolation via Parsortix technology and surface antigen labeling) resulting in a comprehensive workflow. For the CTC-field DanioCTC marks a big step towards functional analysis of CTCs.

We emphasize fact that this is a methodological proof-of-concept study in lines 356-361 the following way:

" In general, DanioCTC provides the methodology to investigate cells with a low prevalence in zebrafish embryos, as shown here with patient derived CTCs. The workflow allows for the selection of cells of interest down to the single-cell level and their subsequent injection. Consequently, the method his paves the way for studies investigating the dissemination of cells in vivo, along with studies focusing on the molecular characterisation of cells after injection."  

2) Indeed, Parsortix is a pre-enrichment system for CTCs and is based on the bigger size and lower deformability of CTCs and not on the expression of biological markers. However, the authors described the CTCs isolated from patients as EpCAM positive and CD45- (there is no description of the method used). Between the pre-enrichment step and the injection of CTCs, tumor cells were then labeled with specific antibodies.

Response: We agree with the reviewer that it is necessary to describe the detection method of CTCs in more detail. We used the EpCAM/CD45 staining to align/harmonize our approach with the FDA-cleared CELLSEARCH system which is considered the gold standard for the enumeration of CTCs. We have added the following description in lines 151-155 (methods section):

"Therefore, enriched MDA-MB-231 cells or CTCs were stained in parallel for EpCAM (1:50, VU1D9, StemcellTM Technologies, Vancouver, Canada, AF488), MUC1 (1:25, CD227, eBioscienceTM, Santa clara, USA, AF488), Her2 (1:50, CellSearch CXC kit, Menarini Group, Florence, Italy, AF488) and CD45 (1:25, 3S-Z5, Santa-Cruz Biotechnology, Dallas, USA, AF647) at 37°C for 60 min for identification by flow cytometry."

3) What was the impact of this staining on the ability of cells to migrate in embryos (this is a key question)?

Response: We agree with the reviewer that this is an important question, which has to be answered for CTCs. There are publications such as from Hamilton et al. [4] showing that CD34-sorted human hematopoietic stem cells home into the zebrafish haematopoietic niche, where they engage with endothelial cells and undergo cell division. This indicates that antibodies may not interfere with cell migration. However, this may depend on the antibody used for labelling. Data from Follain et al. suggest that an antibody against integrins may influence the binding of tumor cells to endothelial cells [5]. It is important to note that CTC xenograft studies in mice (e.g.[6]) made use of anti-EpCAM enriched CTCs for injection. Similarly, Rossi et al. 2013 [7] enriched CTCs with the help of Ferrofluid coupled EpCAM antibodies before their injection into mice. In both studies, EpCAM-positive CTCs retained their migratory capacity. In addition, as our quantifications of injected CTCs in zebrafish embryos at 1 dpi and 3dpi show, up to 87% of initially injected cells can be observed at 3 days post injection, with cells residing in different locations of the host. This suggests only moderate cell loss and the CTCs' ability to disseminate. Altogether, this indicates that cells, upon enrichment with the used antigen-specific antibodies, are viable in zebrafish for several days and that they can move to different locations in the host. Whether active migration is involved in this process and whether enrichment alters their specific attachment capacity, e.g. to endothelial cells, needs to be investigated in dedicated studies in the future.

We have commented on this in the revised version of the manuscript in lines 154/5:

"EpCAM-mediated enrichment of CTCs for xenotransplantation has previously been used [28, 29]."

4) It is definitely mandatory to increase the number of patient samples, especially due to the difference of distribution between MDA & CTC and the variability of their size and general properties.

Response: We agree with the reviewer that it will be desirable to test more patient samples with the newly developed setup. At the moment, we are collecting patient samples with high CTC numbers of different breast cancer types to investigate this question. The intention of the study at hand is to provide a proof-of-concept on the workflow of CTC isolation and injection, which we have emphasized when discussing the limitations of the study in lines 429-433. As stated above, we are convinced that the DanioCTC workflow will be extremely useful for the community to study patient-derived CTCs in vivo.

Lines 429-433:

"Here, we provide a proof-of-concept study on the DanioCTC workflow that allows us to inject isolated CTCs from a single MBC patient into zebrafish embryos, to the best of our knowledge, for the first time. This case study, along with the established setup, will allow for further experiments on isolated CTCs, accompanied by ongoing improvements in the equipment and software."

5) In addition, the use of a single cell line is an issue, and various cell lines (size and origin) should be compared to better understand the effects of their characteristics in the process.

Response: We apologize to the reviewer that the scope of our work has potentially not been clearly stated in the original version of the manuscript. The intention of this study was to develop a workflow for injection of patient-derived CTCs, i.e. of cells that require a specific isolation procedure and are low in abundance, rather than exploring injection of different cell lines into zebrafish. Various cell lines have been transplanted into zebrafish embryos by others, such as MCF-7, BT-474 and MDA-MB-231 in Berens et al. [1]. Thus, the use of MDA-MB-231 cells for xenotransplantation is standard (e.g. [8]) and was used by us to establish the CTCDanio workflow, as this could not be done with the rare and precious CTCs. Furthermore, MDA-MB-231 cells served as a reference in terms of dissemination (possibilities) of tumor cells in zebrafish larvae. After testing MDA-MB-231 cells from culture in the standard setup, MDA-MB-231 cells were spiked into blood samples as a second step, in order to mimic their existence in blood samples and to optimize the cell picking protocol. Only then, as a third step, we tested the workflow for CTCs, since these cells are more delicate to handle than any cultured cells. We explained the fact that MDA-MB-231 cells served as a tool to establish the workflow in lines 303 to 306 the following:

" Since the CTCs were to be isolated from DLA-products obtained from metastatic breast cancer patients, we mimicked such a sample by spiking about 5,000 MDA-MB-231 cells into a CTC-negative 2ml DLA aliquot (negative control) in the next step and processed it according to the workflow depicted in Fig. 1 and Fig. 5."

6) Brief data are shown and many questions arise concerning the survival of cells in the longer term, their organization (cluster formation, mobility,...). A better characterization of the process is mandatory.

Response: We thank the reviewer for this comment. In our work we wish to present a technology, which allows researchers interested in the biology of CTCs to study these cells in vivo in short-term experiments. Such experiments, covering a maximum of 3 days after injection of 2-day old embryos, are feasible under European legislation. Until this age (5 days post fertilization), animal experimentation on zebrafish is exempt from legal regulations, which facilitates experiments in European research laboratories. In the future, it will important to use our setup to enable studies on the biology of CTCs, such as focusing on cluster formation, mobility and migration, extravasation and dormancy. Addressing these topics is absolutely important, but beyond the scope of the (methodological) study at hand. In our study, we developed a novel workflow, which will be the basis to address many interesting questions. It will also allow investigation of longer-term outcomes of CTC xenotransplantation beyond 3 days post injection, provided that animal experimentation beyond day 5 post fertilization is granted.

7) It seems that even with groups of 9 embryos the variability is significant. How to explain this massive variability?

Response: This is a very interesting and important question which likely relates to the variability of CTC gene expression and epigenetic modifications in CTCs. Due to the scarcity of CTCs, we only inject few cells per embryo. In addition, due to the same reason, the sample size with n = 9 is not very large when compared to studies on cell culture derived cancer xenografts (e.g. [9]). Both factors may lead to cell-to-cell variation in terms of dissemination in the host. In the future, it will be of particular interest to correlate cell-to-cell variabilities at the DNA level of CTCs (see Franken et al [10]) with their dissemination behavior. We cautiously discuss this possibility in the revised version of the manuscript the following way in lines 396 to 397:

"In addition, heterogeneity in CTCs might be high at the DNA level."

8) This manuscript shows a system to study CTC in zebrafish but this work offers few original information and requests complementary data.

Response: We hope that the above explanations and revisions to the manuscript addressed the reviewer's concerns and that the DanioCTC methodology can be presented to interested researchers in the form of the provided revised manuscript.

 References in this rebuttal letter

  1. Berens, E.B., et al., Testing the Vascular Invasive Ability of Cancer Cells in Zebrafish (Danio Rerio). J Vis Exp, 2016(117).
  2. Hurtado, P., et al., Modelling metastasis in zebrafish unveils regulatory interactions of cancer-associated fibroblasts with circulating tumour cells. Front Cell Dev Biol, 2023. 11: p. 1076432.
  3. Martínez-Pena, I., et al., Dissecting Breast Cancer Circulating Tumor Cells Competence via Modelling Metastasis in Zebrafish. Int J Mol Sci, 2021. 22(17).
  4. Hamilton, N., I. Sabroe, and S.A. Renshaw, A method for transplantation of human HSCs into zebrafish, to replace humanised murine transplantation models. F1000Res, 2018. 7: p. 594.
  5. Follain, G., et al., Hemodynamic Forces Tune the Arrest, Adhesion, and Extravasation of Circulating Tumor Cells. Dev Cell, 2018. 45(1): p. 33-52.e12.
  6. Baccelli, I., et al., Identification of a population of blood circulating tumor cells from breast cancer patients that initiates metastasis in a xenograft assay. Nat Biotechnol, 2013. 31(6): p. 539-44.
  7. Rossi, E., et al., Retaining the long-survive capacity of Circulating Tumor Cells (CTCs) followed by xeno-transplantation: not only from metastatic cancer of the breast but also of prostate cancer patients. Oncoscience, 2014. 1(1): p. 49-56.
  8. Mercatali, L., et al., Development of a Patient-Derived Xenograft (PDX) of Breast Cancer Bone Metastasis in a Zebrafish Model. Int J Mol Sci, 2016. 17(8).
  9. Tulotta, C., et al., Inhibition of signaling between human CXCR4 and zebrafish ligands by the small molecule IT1t impairs the formation of triple-negative breast cancer early metastases in a zebrafish xenograft model. Dis Model Mech, 2016. 9(2): p. 141-53.
  10. Franken, A., et al., Detection of ESR1 Mutations in Single Circulating Tumor Cells on Estrogen Deprivation Therapy but Not in Primary Tumors from Metastatic Luminal Breast Cancer Patients. J Mol Diagn, 2020. 22(1): p. 111-121.
  11. Stoletov, K., et al., High-resolution imaging of the dynamic tumor cell vascular interface in transparent zebrafish. Proc Natl Acad Sci U S A, 2007. 104(44): p. 17406-11.
  12. Mendelaar, P.A.J., et al., Defining the dimensions of circulating tumor cells in a large series of breast, prostate, colon, and bladder cancer patients. Mol Oncol, 2021. 15(1): p. 116-125.
  13. Fischer, T., A. Hayn, and C.T. Mierke, Effect of Nuclear Stiffness on Cell Mechanics and Migration of Human Breast Cancer Cells. Front Cell Dev Biol, 2020. 8: p. 393.
  14. Dietrich, K., et al., Skeletal Biology and Disease Modeling in Zebrafish. J Bone Miner Res, 2021. 36(3): p. 436-458.

Reviewer 2 Report

Comments and Suggestions for Authors

Reinhardt F., et al present here a manuscript describing an innovative approach to one of the hurdles of use of Zebrafish Xenografts for clinical applications, that is need of high cell numbers for injections in typical needles/ injection set ups.

In summary of this scientific story I would argue the attempt to change the injection set up, enabling collection and transfer of a limited cell number, the methodological part is indeed very interesting and innovative, still the clinical interpretation of the results is not fully convincing. Yes, there are cells detectable after injection, also after several days, but it is not properly discussed if these cells can be qualified as metastasis, can it be really correlated to a phenotype seen in patients? Cells are lost over time, where do they go? Why? Is this of clinical relevance? Do cells simply get stuck, or is it a guided homing, when they end up in head, tail, trunk?

To claim relevance for clinic I need to ask for at least a treatment experiment, can the spreading behavior be influenced for example by compounds? Or would at least different samples behave differently in spreading pattern to retrieve any specific information?

During description of the process of injection in the result part it reads like all collected cells are injected individually after pick up, when I read the methods and out of practical experience I would doubt that the embryo is pricked 50 times. Please clarify/ rephrase this for easy and comprehensive reading.

In the recent years a few original reports were published showing that circulation of xenotransplanted cells is dependent on flow in fish, or size, or markers. I would wish for a discussion of these, because this could correlate your outcome of circulating cells more to biological/ clinical significance.

Author Response

9) … In summary of this scientific story I would argue the attempt to change the injection set up, enabling collection and transfer of a limited cell number, the methodological part is indeed very interesting and innovative, still the clinical interpretation of the results is not fully convincing….

Response: We thank the reviewer for their appreciative comment on the methodology, which is indeed at the core of our work. We agree with the reviewer that it may be currently too far-fetched to relate our data to clinical observations and applications. Indeed, the manuscript at hand is intended to be a proof-of-principle study showing the feasibility of DanioCTC. We hope that our workflow will be taken up by other groups in the near future – since it really solves the issue of low CTC number availability for transplantation into in vivo animal models, a problem that the CTC-field is constantly fighting with.  Employing zebrafish embryos to study CTCs that have been isolated and transplanted with our workflow will lead to the generation of robust CTC-derived models to investigate metastasis, CTC cluster formation, homing, extravasation and cancer cell-to-host cell-interactions. Regarding a clinical use of DanioCTC, we can imagine using the setup for drug testing in the future, for example using CTCs obtained from patients whose CTC number is increasing during metastasis recurrence. This would correspond to a kind of “personalized drug test or screen” and could be combined with CTC DNA analysis. This scenario, however, may be still a long way off.

We have amended the manuscript in terms of relation to clinical observations and clinical use and discuss this topic more cautiously in the revised version of the manuscript (summary statement, lines 77-80, lines 435-437):

"DanioCTC is a novel workflow to inject patient-derived CTCs into zebrafish, enabling studies on the capacity of these rare tumor cells to induce metastases." (summary statement, lines 39-41)

"This approach provides an important model to investigate the CTCs' metastatic potential in zebrafish, addressing one of the requirements to develop personalized therapy approaches and facilitating the discovery of new therapeutic strategies for metastatic cancer treatment." (lines 77-80, introduction)

"Our case study using DanioCTC paves the way for future investigation of MBC patient cohort CTCs to study and characterise the distribution of these cells in vivo." (lines 435-437) 

10) Yes, there are cells detectable after injection, also after several days, but it is not properly discussed if these cells can be qualified as metastasis, can it be really correlated to a phenotype seen in patients?

Response: This is an interesting point that requires more investigation in the future. In particular, longer experiments (beyond day 5 post fertilization/ day 3 post injection) will be required to monitor the fate of CTCs, both in terms of extravasation and their proliferation. As mentioned in the response to question number 6) of reviewer 1, we will have to acquire a specific animal experimentation approval in order to study the zebrafish hosts longer than to 5 days post fertilization. We believe that we would have to grow the zebrafish longer in order to observe cell proliferation. Aside of proliferation and extravasation, which might indicate micrometastasis formation, we envision to re-isolate CTCs which have homed to distinct regions in the zebrafish in order to potentially correlate their genotype with the targeted location in zebrafish. Such future studies will show whether CTC location correlates with accumulation of mutations, for example. At the moment, it is difficult to judge whether the location of CTC in zebrafish correlates in any way with phenotypes in patients. Importantly, injecting tumor cells into the circulation of zebrafish and following up their whereabouts within and outside the circulation is a widely used method to study metastatic properties of tumor cells (e.g. [1]). Here, we make use of a setup that is destined for the injection of rare cell types such as CTCs.

We refer to this topic in the manuscript in lines 392 to 404:

" In contrast, CTCs from a MBC patient injected into zebrafish larvae were primarily located in the head and trunk, unlike MDA-MB-231 cells that were present in the head and tail. The relevance and reasons for this are currently unclear. Flow conditions in the blood vessels of the respective domains, as well as the size or markers of the cells, might influence the outcome [19]. In addition, heterogeneity in CTCs might be high at the DNA level [36]. However, caution is warranted as our observation is based on a single experiment and DLA aliquot. Therefore, further investigation is required to determine if there is a correlation between brain metastases and localization of CTCs to the head in zebrafish embryos. In the future, it will be important to isolate tumor cells from different regions of injected zebrafish embryos to compare their gene expression profiles with CTCs and DTCs from the patient to examine their expression profiles and clonal relationship. Furthermore, more MBC patient derived samples should be investigated with the DanioCTC workflow.

11) Cells are lost over time, where do they go? Why? Is this of clinical relevance? Do cells simply get stuck, or is it a guided homing, when they end up in head, tail, trunk?

Response: These are very interesting questions which have to be addressed in the future. At the moment, trying to answer these questions would, unfortunately, be rather speculative. We know from work with cell lines that cancer cells do not simply get stuck in the vasculature of zebrafish. Instead, they are able to move within the circulation and can adhere to endothelial cells (Follain et al. [5], Stoletov et al. [11]). In addition, these cells can extravasate into the surrounding tissues. It is also clear that a fraction of the injected cells does not survive long-term. Whether CTCs home to specific sites in our model, are able to extravasate and proliferate is currently unclear, and needs to be investigated in the future. The fact that up to 87% of initially injected cells can be observed at 3 days post injection, suggests that the majority of cells is viable in xenotransplanted zebafish. With DanioCTC we hope to stir interest in this topic by opening up a method to address these questions.

In the revised version of the manuscript, we refer to these uncertainties and future avenues in the following way (lines 394-399):

" The relevance and reasons for this are currently unclear. Flow conditions in the blood vessels of the respective domains, as well as the size or markers of the cells, might influence the outcome [19]. In addition, heterogeneity in CTCs might be high at the DNA level [36]. However, caution is warranted as our observation is based on a single experiment and DLA aliquot. Therefore, further investigation is required... "

12) To claim relevance for clinic I need to ask for at least a treatment experiment, can the spreading behavior be influenced for example by compounds? Or would at least different samples behave differently in spreading pattern to retrieve any specific information?

Response: We agree with the reviewer that it is too early to claim relevance for clinical use. Instead, the workflow has to be used in the future to explore use in a clinical setting. Due to limitations in sample material, we were not yet able to test the patient's CTC response to clinically used drugs. Thus, we agree that there should be less emphasis on clinical relevance of DanioCTC and have changed the manuscript accordingly (see also response to question 9).

"DanioCTC is a novel workflow to inject patient-derived CTCs into zebrafish, enabling studies on the capacity of these rare tumor cells to induce metastases." (summary statement, lines 39-41)

"This approach provides an important model to investigate the CTCs' metastatic potential in zebrafish, addressing one of the requirements to develop personalized therapy approaches and facilitating the discovery of new therapeutic strategies for metastatic cancer treatment." (lines 77-80, introduction)

"Our case study using DanioCTC paves the way for future investigation of MBC patient cohort CTCs to study and characterise the distribution of these cells in vivo. " (lines 435-37) 

13) During description of the process of injection in the result part it reads like all collected cells are injected individually after pick up, when I read the methods and out of practical experience I would doubt that the embryo is pricked 50 times. Please clarify/ rephrase this for easy and comprehensive reading.

Response: Thank you very much for this comment. We are sorry that this description has been confusing. We have been more explicit in the revised version of the manuscript (lines 187-189 & lines 216-223):

"Approximately 5 – 10 nl of the capillary volume, containing the settled 50 picked cells in the capillary tip, were semi-automatically micro-injected with the CellCelector into the DoC of individual zebrafish." (lines 187-189 )

" Development of a CTC injection workflow - concept and challenges

We set up an injection workflow for low CTC numbers by combining, adapting and fully integrating existing, well-validated workflows for each of the required technology modules: (i) xenotransplantation of cancer cells into zebrafish embryos by microinjection into the blood circulation; (ii) DLA, which allows to process large volumes of blood for CTC analysis  (iii) flow cytometry for the provision of well-characterized CTC preparations; and (iv) the CellCelector automated micromanipulation system ¬– extended by an additional optical system – for isolation of individual cells and performing their collective injection. " (lines 216-223)

14) In the recent years a few original reports were published showing that circulation of xenotransplanted cells is dependent on flow in fish, or size, or markers. I would wish for a discussion of these, because this could correlate your outcome of circulating cells more to biological/ clinical significance.

Response: Thanks very much for this suggestion. Indeed, cell lines and also CTCs differ in physico-mechanical features such as size or stiffness [12, 13] which may represent an explanation for the differences in ‘homing’ we observe between MDA-MB 231 cells and CTCs. This is also the reason why we developed a workflow avoiding the use of ferrofluid or magnetic particles for CTC capture as used by the CELLSEARCH system. It is also true that flow in xenotransplanted zebrafish has been shown to be decisive in arrest, adhesion and extravasation of xenotransplanted tumor cells in the circulation (e.g. Follain et al. [5]). Moreover, surface antigen composition will impact on the capacity of cancer cells to adhere to and passage blood vessels (overview in Dietrich et al. 2021 [14]). Indeed, it is very likely that the three factors flow, CTC size and CTC expression impact on the CTCs survival and spread in the circulation and beyond; however, we have not investigated this. We discuss the possibility that the dissemination of CTCs to specific sites might be impacted by these factors in the discussion of the manuscript (lines 371 to 377):

" In this transparent model, the capacity of CTCs to arrest and adhere to the endothelium, which may involve cell rolling and other processes that can be observed in tumor cells and immune cells found in the circulation [33] can be assessed in a dynamic fashion, e.g. by high resolution video microscopy. In addition, hemodynamic forces of blood flow that impact the arrest of CTCs can be studied, e.g. by monitoring different locations in the embryo displaying distinct flow rates, and manipulating flow rates by heartbeat-modifying drugs [34].”

References in this rebuttal letter

  1. Berens, E.B., et al., Testing the Vascular Invasive Ability of Cancer Cells in Zebrafish (Danio Rerio). J Vis Exp, 2016(117).
  2. Hurtado, P., et al., Modelling metastasis in zebrafish unveils regulatory interactions of cancer-associated fibroblasts with circulating tumour cells. Front Cell Dev Biol, 2023. 11: p. 1076432.
  3. Martínez-Pena, I., et al., Dissecting Breast Cancer Circulating Tumor Cells Competence via Modelling Metastasis in Zebrafish. Int J Mol Sci, 2021. 22(17).
  4. Hamilton, N., I. Sabroe, and S.A. Renshaw, A method for transplantation of human HSCs into zebrafish, to replace humanised murine transplantation models. F1000Res, 2018. 7: p. 594.
  5. Follain, G., et al., Hemodynamic Forces Tune the Arrest, Adhesion, and Extravasation of Circulating Tumor Cells. Dev Cell, 2018. 45(1): p. 33-52.e12.
  6. Baccelli, I., et al., Identification of a population of blood circulating tumor cells from breast cancer patients that initiates metastasis in a xenograft assay. Nat Biotechnol, 2013. 31(6): p. 539-44.
  7. Rossi, E., et al., Retaining the long-survive capacity of Circulating Tumor Cells (CTCs) followed by xeno-transplantation: not only from metastatic cancer of the breast but also of prostate cancer patients. Oncoscience, 2014. 1(1): p. 49-56.
  8. Mercatali, L., et al., Development of a Patient-Derived Xenograft (PDX) of Breast Cancer Bone Metastasis in a Zebrafish Model. Int J Mol Sci, 2016. 17(8).
  9. Tulotta, C., et al., Inhibition of signaling between human CXCR4 and zebrafish ligands by the small molecule IT1t impairs the formation of triple-negative breast cancer early metastases in a zebrafish xenograft model. Dis Model Mech, 2016. 9(2): p. 141-53.
  10. Franken, A., et al., Detection of ESR1 Mutations in Single Circulating Tumor Cells on Estrogen Deprivation Therapy but Not in Primary Tumors from Metastatic Luminal Breast Cancer Patients. J Mol Diagn, 2020. 22(1): p. 111-121.
  11. Stoletov, K., et al., High-resolution imaging of the dynamic tumor cell vascular interface in transparent zebrafish. Proc Natl Acad Sci U S A, 2007. 104(44): p. 17406-11.
  12. Mendelaar, P.A.J., et al., Defining the dimensions of circulating tumor cells in a large series of breast, prostate, colon, and bladder cancer patients. Mol Oncol, 2021. 15(1): p. 116-125.
  13. Fischer, T., A. Hayn, and C.T. Mierke, Effect of Nuclear Stiffness on Cell Mechanics and Migration of Human Breast Cancer Cells. Front Cell Dev Biol, 2020. 8: p. 393.
  14. Dietrich, K., et al., Skeletal Biology and Disease Modeling in Zebrafish. J Bone Miner Res, 2021. 36(3): p. 436-458.

Reviewer 3 Report

Comments and Suggestions for Authors

Reinhardt et al. presented the manuscript describe that DanioCTC is an innovative xenograft workflow that overcomes the scarcity of patient-derived CTCs in animal models. Authors combining DLA, the Parsortix microfluidic system, flow cytometry, and the CellCelector setup, DanioCTC effectively enriches and isolates CTCs from MBC patients for injection into zebrafish embryos. The manuscript was carried out carefully and has completed the writing describes the techniques and findings clearly. Authors have established an innovative workflow for the injection of MBC- derived isolated CTCs into zebrafish embryos, enabling live imaging of patient-derived CTC dissemination in xenografts over time. The manuscripts provide a new method and techniques for studies the biology of metastatic breast cancer and developing targeted interventions. Therefore, the manuscript is sufficient to publish in the journal.

Author Response

15) Reinhardt et al. presented the manuscript describe that DanioCTC is an innovative xenograft workflow that overcomes the scarcity of patient-derived CTCs in animal models. Authors combining DLA, the Parsortix microfluidic system, flow cytometry, and the CellCelector setup, DanioCTC effectively enriches and isolates CTCs from MBC patients for injection into zebrafish embryos. The manuscript was carried out carefully and has completed the writing describes the techniques and findings clearly. Authors have established an innovative workflow for the injection of MBC- derived isolated CTCs into zebrafish embryos, enabling live imaging of patient-derived CTC dissemination in xenografts over time. The manuscripts provide a new method and techniques for studies the biology of metastatic breast cancer and developing targeted interventions. Therefore, the manuscript is sufficient to publish in the journal.

Response: We are happy that the manuscript meets the reviewer's expectation and would like to thank the reviewer for her/his positive assessment.

Reviewer 4 Report

Comments and Suggestions for Authors

The manuscript “Analysis of circulating tumor cells from metastatic breast cancer patients in zebrafish xenografts” reports on an approach enabling the transfer of human circulating tumour cells to the circulatory system of zebrafish embryos and the quantification of their distribution and survival over a two-day period. Zebrafish embryos allow for transplantation and observation of small numbers of cells per embryo, so they are particularly suitable for in vivo studies of cells that are difficult to obtain in large numbers. I therefore find that such methodology has useful applications.

I have some comments on the manuscript that the authors might wish to consider.

Simple summary / Abstract:

The final sentence of the Simple summary is adhering to the Abstract. These sections need to be clearly separated.

It would help readers if the abbreviation “MBC” in the Abstract were explained.

Line 34: when MDA-MB-231 cells spiked into DLA aliquots were processed… - A person only reading the abstract might not understand what this means. I think the abstract is completely OK without this sentence, but if the authors find this information critical, they might wish to rephrase it so that anyone can understand what was done.

Introduction:

Line 57: This approach provides a clinically relevant setting… - I suggest the use of caution when stating that zebrafish embryos are a clinically relevant setting.

Methods:

Line 87: spike in experiments – for readers unfamiliar with this methodology, a very brief statement of what “spike-in” means in this context would be useful.

Line 98: It might be useful to explain why eGFP-labelled cells were additionally stained with SP-DiOC18. I was perplexed when reading this. Then I read the entire manuscript several times and was still perplexed.

Line 106: The salts comprising E3 medium are listed in German, I suggest that they are written out in English.

Line 105: a petri dish containing 1.5% low melting agarose – What is not clear to me is whether the embryos were embedded in this agarose solution or it only covered the bottom of the petri dish with the embryos on top of the agarose.

Line 140: 100 mbar of pressure – I am guessing this is 100 mbar above ambient pressure.

Line 160: Cell suspensions were transferred onto glass slides – Please, specify the type and manufacturer of slides.

Line 164: 200 ms (FITC) and 200 ms (Cy5) – As the exposure times are identical, only writing out one would be enough.

Line 179: Zebrafish embryos were imaged at 1 dpi and 3 dpi – In my opinion, two important bits of information are missing in the description of the protocol. What happened to the embryos that were selected for further imaging? I suppose they were moved to some new container with anaesthetic-free medium. Secondly, it is unclear from the text how individual embryos were identified for re-imaging.

A small issue I have with this approach is the maintenance of embryos. If all of them were kept in the same petri dish, I am wondering if these can be considered independent measurements for further analysis.

Line 192: were isolated of a patient – I suggest from a patient

Line 197: Repeated Measures ANOVA followed by Bonferroni’s tests – I suppose it is meant that pairwise comparisons were performed with paired measurements t-tests using the Bonferroni correction.

I have a minor issue with the statistical analysis in this case. Comparisons were not only made between day 1 and day 3, for which repeated measures tests such as the Friedman test and repeated measures ANOVA are suitable, but also between body regions, for which they are not. When body regions are compared at a single time-point, ordinary two-way ANOVA / t-test or Mann-Whitney / Kruskal-Wallis tests would be appropriate, with suitable corrections for the number of comparisons.

Results:

Throughout the Results, much of the text describes experimental procedures, which belongs to the Methods section. Some of this text might improve the Methods section if it were moved there and removed from the Results.

3.2. Adaptations that were required to realize DanioCTC: Titles of subsections

Each paragraph here begins with its own title that ends with a colon. I think these titles could either be omitted or be written as subsection titles in separate lines.

Figure 1: Please make sure that you have the rights to publish this particular image of the CellCelector device.

Figures 6 and S4: Since no significance levels are provided on the image, the caption might include a comment that no significant differences between groups were observed.

Author Response

16) The manuscript “Analysis of circulating tumor cells from metastatic breast cancer patients in zebrafish xenografts” reports on an approach enabling the transfer of human circulating tumour cells to the circulatory system of zebrafish embryos and the quantification of their distribution and survival over a two-day period. Zebrafish embryos allow for transplantation and observation of small numbers of cells per embryo, so they are particularly suitable for in vivo studies of cells that are difficult to obtain in large numbers. I therefore find that such methodology has useful applications.

Response: We thank the reviewer for their appreciative assessment.

I have some comments on the manuscript that the authors might wish to consider.

Simple summary / Abstract:

17) The final sentence of the Simple summary is adhering to the Abstract. These sections need to be clearly separated.

Response: We thank the reviewer for pointing out this error to us. We have changed the summary statement in lines 39-41 to the following:

"DanioCTC is a novel workflow to inject patient-derived CTCs into zebrafish, enabling studies on the capacity of these rare tumor cells to induce metastases."

18) It would help readers if the abbreviation “MBC” in the Abstract were explained.

Response: We have explained the abbreviation "MBC" accordingly in line 31:

"... DanioCTC effectively enriches and isolates CTCs from metastatic breast cancer (MBC) patients for injection into zebrafish embryos."

19) Line 31: when MDA-MB-231 cells spiked into DLA aliquots were processed… - A person only reading the abstract might not understand what this means. I think the abstract is completely OK without this sentence, but if the authors find this information critical, they might wish to rephrase it so that anyone can understand what was done.

Response: We thank the reviewer for pointing this out. We now explain the term ‘spike in experiment’ in this context in the methods section (lines 139-143) the following way:

"For spike in experiments, MDA-MB-231 cell line cells were added into diluted negative control DLA samples. Therefore, cultured MDA-MB-231 cells were trypsinized using trypsin–EDTA (0.05%), neutralized using complete medium, centrifuged at 1,200 rpm for 5 min and re-suspended in PBS. Subsequently, 5,000 MDA-MB-231 cell line cells were transferred into diluted negative control DLA samples".

In addition, we added information on the term "spiked" in the abstract (lines 33 to 35):

"Notably, when MDA-MB-231 cells spiked (i.e. supplemented) into DLA aliquots were processed using DanioCTC, cell dissemination patterns remained consistent. "

Introduction:

20) Line 57: This approach provides a clinically relevant setting… - I suggest the use of caution when stating that zebrafish embryos are a clinically relevant setting.

Response: We agree with the reviewer that it is probably precocious to propose clinical relevance of DanioCTC. In agreement with comments 9) and 12) (both Reviewer 2) we have amended the manuscript in a way so that the potential clinical utility of DanioCT is stated more cautiously. Some examples of these more cautious explanations are found below (summary statement lines 39/41, line 77, line 429):

"DanioCTC is a novel workflow to inject patient-derived CTCs into zebrafish, enabling studies on the capacity of these rare tumor cells to induce metastases." (summary statement, lines 39-41)

"This approach provides an important model to investigate the CTCs' metastatic potential in zebrafish, addressing one of the requirements to develop personalized therapy approaches and facilitating the discovery of new therapeutic strategies for metastatic cancer treatment." (lines 77-80, introduction)

"Our case study using DanioCTC paves the way for future investigation of MBC patient cohort CTCs to study and characterise the distribution of these cells in vivo. " (lines 435-37) 

Methods:

21) Line 87: spike in experiments – for readers unfamiliar with this methodology, a very brief statement of what “spike-in” means in this context would be useful.

Response: We now explain the term ‘spike in experiment’ in this context in the methods section (lines 139-143) the following way (please see also point 19):

"For spike in experiments, MDA-MB-231 cell line cells were added into diluted negative control DLA samples. Therefore, cultured MDA-MB-231 cells were trypsinized using trypsin–EDTA (0.05%), neutralized using complete medium, centrifuged at 1,200 rpm for 5 min and re-suspended in PBS. Subsequently, 5,000 MDA-MB-231 cell line cells were transferred into diluted negative control DLA samples."

Furthermore, we explain spike in experiments in lines 105-107: "Cultured cells were harvested at a confluence of approximately 80% for staining and spike in experiments (i.e. experiments in which tumor cell-free samples were supplemented with tumor cells)."

In addition, we added information on the term "spiked" in the abstract (lines 33 to 35):

"Notably, when MDA-MB-231 cells spiked (i.e. supplemented) into DLA aliquots were processed using DanioCTC, cell dissemination patterns remained consistent."

22) Line 98: It might be useful to explain why eGFP-labelled cells were additionally stained with SP-DiOC18. I was perplexed when reading this. Then I read the entire manuscript several times and was still perplexed.

Response: Indeed, this sounds counterintuitive. Thus, we have added an explanation in the methods section (lines 101-105):

"EGFP labeled MDA-MB-231 cells were trypsinized using trypsin–EDTA (0.25%), neutralized using complete medium, centrifuged at 1,200 rpm for 10 min, re-suspended in PBS, and additionally stained with SP-DiOC18(3) (D7778, Invitrogen) according to the manufacturer's recommendations, in order to also stain isolated cells that might have lost their GFP fluorescence (clonal vector loss) during cell division."  

23) Line 106: The salts comprising E3 medium are listed in German, I suggest that they are written out in English.

Response: We apologize for this mistake. It was corrected in the revised version of the manuscript (lines 110-114) to the following:

"Zebrafish embryos (2 days post fertilization, dpf) were manually dechorionated, anesthetized using 0.02% tricaine and transferred into a petri dish which was covered with a thin layer of 1.5% low melting agarose in E3 (50 mM sodium chloride, 0.17 mM potassium chloride, 0.33 mM calcium chloride, 0.33 mM magnesium sulfate), in order to avoid adhesion of the embryos to the dish."

24) Line 105: a petri dish containing 1.5% low melting agarose – What is not clear to me is whether the embryos were embedded in this agarose solution or it only covered the bottom of the petri dish with the embryos on top of the agarose.

Response: We clarified this in the methods section - the dish is covered by agarose so that embryos do not get damaged; embryos are not embedded in the agarose.

"Zebrafish embryos (2 days post fertilization, dpf) were manually dechorionated, anesthetized using 0.02% tricaine and transferred into a petri dish which was covered with a thin layer of 1.5% low melting agarose in E3 (50 mM sodium chloride, 0.17 mM potassium chloride, 0.33 mM calcium chloride, 0.33 mM magnesium sulfate), in order to avoid adhesion of the embryos to the dish." (lines 110-114)

25) Line 140: 100 mbar of pressure – I am guessing this is 100 mbar above ambient pressure.

Response: The reviewer is correct - we have adapted this information accordingly (lines 146 to 147):

"Following the protocol, filtration cassettes with 6.5 µm gaps were used and 100 mbar above ambient pressure were applied"

26) Line 160: Cell suspensions were transferred onto glass slides – Please, specify the type and manufacturer of slides.

Response: We have added the required information (lines 170-171):

"Cell suspensions were transferred onto ALS MagnetPick glass slides (Sartorius, CC0059), placed on the automatic stage of the CellCelector microscope and were allowed to settle. "

27) Line 164: 200 ms (FITC) and 200 ms (Cy5) – As the exposure times are identical, only writing out one would be enough.

Response: We changed this as per the reviewer's suggestion (lines 174/175):

"The following exposure time was used: 200 ms (FITC and Cy5)."

28) Line 179: Zebrafish embryos were imaged at 1 dpi and 3 dpi – In my opinion, two important bits of information are missing in the description of the protocol. What happened to the embryos that were selected for further imaging? I suppose they were moved to some new container with anaesthetic-free medium. Secondly, it is unclear from the text how individual embryos were identified for re-imaging.

Response: We thank the reviewer for pointing out this imprecision. In the revised version of the manuscript, we have provided the respective information in lines 189-193) in the following way:

"Engrafted embryos were maintained in a new petri dish filled with E3 medium at 34°C. Based on the fluorescence spread of the injected embryos at 2 h post injection (hpi), embryos with tumor cells in the blood circulation were selected and each xenografted zebrafish embryo was transferred into a single dish filled with E3 medium for maintenance for up to 3 dpi."

29) A small issue I have with this approach is the maintenance of embryos. If all of them were kept in the same petri dish, I am wondering if these can be considered independent measurements for further analysis.

Response: Zebrafish embryos were kept in single dishes for identification and in between imaging. Please also see point 29).

"Engrafted embryos were maintained in a new petri dish filled with E3 medium at 34°C. Based on the fluorescence spread of the injected embryos at 2 h post injection (hpi), embryos with tumor cells in the blood circulation were selected and each xenografted zebrafish embryo was transferred into a single dish filled with E3 medium for maintenance for up to 3 dpi." (lines 189-193)

30) Line 192: were isolated of a patient – I suggest from a patient

Response: We adapted the text accordingly (lines 205-207 of the revised version of the manuscript):

"Injected CTCs were isolated from a 57-year-old patient with a hormone-receptor positive, Her2/neu negative breast cancer with bone, bone marrow and lymph node metastases."

31) Line 197: Repeated Measures ANOVA followed by Bonferroni’s tests – I suppose it is meant that pairwise comparisons were performed with paired measurements t-tests using the Bonferroni correction.

Reply: Yes, a Bonferroni post hoc test as a series of independent t tests were performed on each pair of groups. Please also refer to point 32).

32) I have a minor issue with the statistical analysis in this case. Comparisons were not only made between day 1 and day 3, for which repeated measures tests such as the Friedman test and repeated measures ANOVA are suitable, but also between body regions, for which they are not. When body regions are compared at a single time-point, ordinary two-way ANOVA / t-test or Mann-Whitney / Kruskal-Wallis tests would be appropriate, with suitable corrections for the number of comparisons.

Response: We thank the reviewer for providing advice in terms of statistical analysis. As per the reviewer's suggestion, we have re-evaluated with ANOVA followed by a post-hoc Bonferroni test for parametric data or Kuskal-Wallis test and a post-hoc Dunn's test for non-parametric data. The statistical tests are listed in the methods section and in the figure captions. The graphs have been adapted accordingly.

"Statistical analyses were performed for parametric data by ANOVA followed by post-hoc Bonferroni test. For non-parametric data, statistical analyses were performed by Kruskal-Wallis test followed by post-hoc Dunn's test, in order to correct for multiple comparisons. (lines 210-213, methods)

" ANOVA followed by post-hoc Bonferroni test" (captions Figure 4 and Supplement Figure 2)

" Kruskal-Wallis test followed by post-hoc Dunn's test" (captions Figures 5, 6 and Supplement Figures 3,4)

Results:

33) Throughout the Results, much of the text describes experimental procedures, which belongs to the Methods section. Some of this text might improve the Methods section if it were moved there and removed from the Results.

Response: The DanioCTC concept, challenges and adaptations partly reflect the results of our DanioCTC development and are supposed to guide the reader through the procedure when using the workflow. Nevertheless, as per the reviewer's suggestion, we removed and moved technical details on the standard xenotransplantation and CTC injection to the method section. For example, paragraph line 289 following and paragraph line 292 following was removed in the results section and line 176 in the methods section adapted.

34) Each paragraph here begins with its own title that ends with a colon. I think these titles could either be omitted or be written as subsection titles in separate lines.

Response: We thank the reviewer for this suggestion. We adapted the respective phrases to subsection titles.

35) Figure 1: Please make sure that you have the rights to publish this particular image of the CellCelector device.

Response: The owner of this image and the developer of the CellCelector, Jens Eberhardt, is co-author. He provided the image.

36) Figures 6 and S4: Since no significance levels are provided on the image, the caption might include a comment that no significant differences between groups were observed.

Response: As per suggestion, we have added a sentence to the caption of Figure 6.

"No statistically significant differences were observed between groups." (Caption of Figure 6)

References in this rebuttal letter

  1. Berens, E.B., et al., Testing the Vascular Invasive Ability of Cancer Cells in Zebrafish (Danio Rerio). J Vis Exp, 2016(117).
  2. Hurtado, P., et al., Modelling metastasis in zebrafish unveils regulatory interactions of cancer-associated fibroblasts with circulating tumour cells. Front Cell Dev Biol, 2023. 11: p. 1076432.
  3. Martínez-Pena, I., et al., Dissecting Breast Cancer Circulating Tumor Cells Competence via Modelling Metastasis in Zebrafish. Int J Mol Sci, 2021. 22(17).
  4. Hamilton, N., I. Sabroe, and S.A. Renshaw, A method for transplantation of human HSCs into zebrafish, to replace humanised murine transplantation models. F1000Res, 2018. 7: p. 594.
  5. Follain, G., et al., Hemodynamic Forces Tune the Arrest, Adhesion, and Extravasation of Circulating Tumor Cells. Dev Cell, 2018. 45(1): p. 33-52.e12.
  6. Baccelli, I., et al., Identification of a population of blood circulating tumor cells from breast cancer patients that initiates metastasis in a xenograft assay. Nat Biotechnol, 2013. 31(6): p. 539-44.
  7. Rossi, E., et al., Retaining the long-survive capacity of Circulating Tumor Cells (CTCs) followed by xeno-transplantation: not only from metastatic cancer of the breast but also of prostate cancer patients. Oncoscience, 2014. 1(1): p. 49-56.
  8. Mercatali, L., et al., Development of a Patient-Derived Xenograft (PDX) of Breast Cancer Bone Metastasis in a Zebrafish Model. Int J Mol Sci, 2016. 17(8).
  9. Tulotta, C., et al., Inhibition of signaling between human CXCR4 and zebrafish ligands by the small molecule IT1t impairs the formation of triple-negative breast cancer early metastases in a zebrafish xenograft model. Dis Model Mech, 2016. 9(2): p. 141-53.
  10. Franken, A., et al., Detection of ESR1 Mutations in Single Circulating Tumor Cells on Estrogen Deprivation Therapy but Not in Primary Tumors from Metastatic Luminal Breast Cancer Patients. J Mol Diagn, 2020. 22(1): p. 111-121.
  11. Stoletov, K., et al., High-resolution imaging of the dynamic tumor cell vascular interface in transparent zebrafish. Proc Natl Acad Sci U S A, 2007. 104(44): p. 17406-11.
  12. Mendelaar, P.A.J., et al., Defining the dimensions of circulating tumor cells in a large series of breast, prostate, colon, and bladder cancer patients. Mol Oncol, 2021. 15(1): p. 116-125.
  13. Fischer, T., A. Hayn, and C.T. Mierke, Effect of Nuclear Stiffness on Cell Mechanics and Migration of Human Breast Cancer Cells. Front Cell Dev Biol, 2020. 8: p. 393.
  14. Dietrich, K., et al., Skeletal Biology and Disease Modeling in Zebrafish. J Bone Miner Res, 2021. 36(3): p. 436-458.

Reviewer 5 Report

Comments and Suggestions for Authors

In the present manuscript, Reinhardt et al. established an innovative method (DanioCTC) to monitor and study how cancer spread by using circulating tumor cells (CTCs) in zebrafish models.

The authors combined several advanced technologies to enrich and isolate CTCs from patients and injected these isolated CTCs into zebrafish embryos then studied the dissemination patterns. They found that CTCs from patients were primarily localized in the head and trunk regions of the zebrafish embryos.

Overall, DanioCTC seems to be a novel workflow for better understanding the biology of metastatic breast cancer and developing targeted interventions. However, there are some questions that still need to be clarified.

1.      Patient’ CTCs were primarily localized in the head and trunk regions but MDA-MB-231 cells were primarily localized in the head and tail regions. Patients in this study were HR+/Her2-, but MDA-MB-231 cell line is a TNBC subtype. Authors should discuss this difference.

2.      Although DanioCTC could be used as a tool in the CTC study, however, how to perform it practically in clinical testing?

3.      How to study the hemodynamic forces of blood flow on impacting the arrest of CTCs by DanioCTC?

Comments on the Quality of English Language

none.

Author Response

36) Overall, DanioCTC seems to be a novel workflow for better understanding the biology of metastatic breast cancer and developing targeted interventions. However, there are some questions that still need to be clarified.

Response: Many thanks for reviewing our manuscript and providing constructive feedback.

37) Patient’ CTCs were primarily localized in the head and trunk regions but MDA-MB-231 cells were primarily localized in the head and tail regions. Patients in this study were HR+/Her2-, but MDA-MB-231 cell line is a TNBC subtype. Authors should discuss this difference.

Response: This is an important point. In our study, we have used the MDA-MB-231 cell line as a tool to establish our workflow, which is intended as a proof-of-concept study demonstrating that patient-derived CTCs can be isolated and xenotransplanted with the help of our system. In the future, it will be important to test more CTC samples to characterize CTC distribution in more detail and then cell lines with more expression similarities could be used to compare dissemination. We wish to be not too speculative about the differences between CTC and MDA-MB-231 cell dissemiation, but are briefly touching on this topic in lines 392 to 404 of the revised manuscript:

"In contrast, CTCs from a MBC patient injected into zebrafish larvae were primarily located in the head and trunk, unlike MDA-MB-231 cells that were present in the head and tail. The relevance and reasons for this are currently unclear. ... Therefore, further investigation is required to determine if there is a correlation between brain metastases and localization of CTCs to the head in zebrafish embryos. ... . Furthermore, more MBC patient derived samples should be investigated with the DanioCTC workflow."

38) Although DanioCTC could be used as a tool in the CTC study, however, how to perform it practically in clinical testing?

Reply: We thank the reviewer for this question. The performed proof-of-concept study aims at providing the basis for the investigation of patient-derived CTCs and could therefore be developed further to be used in a clinical setting. At the current stage, this is not yet established, and the setup in its current state is primarily useful to address fundamental questions on the biology of CTCs. DanioCTC is an additional piece in the toolbox to better understand the puzzling cell type of CTCs . Once the biology is understood, there is indeed translational potential, such as by testing drugs on CTCs obtained from patients whose CTC number is increasing during metastasis recurrence. As our current setup is not yet intended for clinical use, we have decreased the emphasis on clinical relevance of DanioCTC in the manuscript. We now discuss this topic more cautiously in the revised version of the manuscript (please also see point 9) of reviewer 2):

"DanioCTC is a novel workflow to inject patient-derived CTCs into zebrafish, enabling studies on the capacity of these rare tumor cells to induce metastases." (summary statement, lines 39-41)

"This approach provides an important model to investigate the CTCs' metastatic potential in zebrafish, addressing one of the requirements to develop personalized therapy approaches and facilitating the discovery of new therapeutic strategies for metastatic cancer treatment." (lines 77-80, introduction)

"Our case study using DanioCTC paves the way for future investigation of MBC patient cohort CTCs to study and characterise the distribution of these cells in vivo. " (lines 435-37) 

  1. How to study the hemodynamic forces of blood flow on impacting the arrest of CTCs by DanioCTC?

Response: In the past, it has been shown that hemodynamic forces and blood flow are decisive for the arrest, adhesion and extravasation of xenotransplanted tumor cells in zebrafish (e.g. Follain et al. [5]). In order to study hemodynamic forces of blood flow, DanioCTC could be used for the injection of CTCs into zebrafish embryos and follow up analyses, such as assessing different positions in the zebrafish circulation (with different flow rates), manipulating the organism's blood flow by drugs (to decrease or increase the heartbeat rate) or using mutants with altered blood flow, as previously shown for cell culture derived tumor cells. We relate to the possibility of assessing CTC arrest with our system in the future in lines 371 to 377 of the revised version of the manuscript (please also see point 14) reviewer 2):

"In this transparent model, the capacity of CTCs to arrest and adhere to the endothelium, which may involve cell rolling and other processes that can be observed in tumor cells and immune cells found in the circulation [33] can be assessed in a dynamic fashion, e.g. by high resolution video microscopy. In addition, hemodynamic forces of blood flow that impact the arrest of CTCs can be studied, e.g. by monitoring different locations in the embryo displaying distinct flow rates, and manipulating flow rates by heartbeat-modifying drugs [34]."

We thank all reviewers for their comments on our manuscript, which we hope now meets the quality criteria for publication in Cancers.

References in this rebuttal letter

  1. Berens, E.B., et al., Testing the Vascular Invasive Ability of Cancer Cells in Zebrafish (Danio Rerio). J Vis Exp, 2016(117).
  2. Hurtado, P., et al., Modelling metastasis in zebrafish unveils regulatory interactions of cancer-associated fibroblasts with circulating tumour cells. Front Cell Dev Biol, 2023. 11: p. 1076432.
  3. Martínez-Pena, I., et al., Dissecting Breast Cancer Circulating Tumor Cells Competence via Modelling Metastasis in Zebrafish. Int J Mol Sci, 2021. 22(17).
  4. Hamilton, N., I. Sabroe, and S.A. Renshaw, A method for transplantation of human HSCs into zebrafish, to replace humanised murine transplantation models. F1000Res, 2018. 7: p. 594.
  5. Follain, G., et al., Hemodynamic Forces Tune the Arrest, Adhesion, and Extravasation of Circulating Tumor Cells. Dev Cell, 2018. 45(1): p. 33-52.e12.
  6. Baccelli, I., et al., Identification of a population of blood circulating tumor cells from breast cancer patients that initiates metastasis in a xenograft assay. Nat Biotechnol, 2013. 31(6): p. 539-44.
  7. Rossi, E., et al., Retaining the long-survive capacity of Circulating Tumor Cells (CTCs) followed by xeno-transplantation: not only from metastatic cancer of the breast but also of prostate cancer patients. Oncoscience, 2014. 1(1): p. 49-56.
  8. Mercatali, L., et al., Development of a Patient-Derived Xenograft (PDX) of Breast Cancer Bone Metastasis in a Zebrafish Model. Int J Mol Sci, 2016. 17(8).
  9. Tulotta, C., et al., Inhibition of signaling between human CXCR4 and zebrafish ligands by the small molecule IT1t impairs the formation of triple-negative breast cancer early metastases in a zebrafish xenograft model. Dis Model Mech, 2016. 9(2): p. 141-53.
  10. Franken, A., et al., Detection of ESR1 Mutations in Single Circulating Tumor Cells on Estrogen Deprivation Therapy but Not in Primary Tumors from Metastatic Luminal Breast Cancer Patients. J Mol Diagn, 2020. 22(1): p. 111-121.
  11. Stoletov, K., et al., High-resolution imaging of the dynamic tumor cell vascular interface in transparent zebrafish. Proc Natl Acad Sci U S A, 2007. 104(44): p. 17406-11.
  12. Mendelaar, P.A.J., et al., Defining the dimensions of circulating tumor cells in a large series of breast, prostate, colon, and bladder cancer patients. Mol Oncol, 2021. 15(1): p. 116-125.
  13. Fischer, T., A. Hayn, and C.T. Mierke, Effect of Nuclear Stiffness on Cell Mechanics and Migration of Human Breast Cancer Cells. Front Cell Dev Biol, 2020. 8: p. 393.
  14. Dietrich, K., et al., Skeletal Biology and Disease Modeling in Zebrafish. J Bone Miner Res, 2021. 36(3): p. 436-458.

Round 2

Reviewer 2 Report

Comments and Suggestions for Authors

Dear authors, thank you for discussing my comments to length, I appreciate that you found my questions interesting. In my regard for Zebrafish tumor cell xenografts I would say it is unfortunate that you have missed the chance to provide more experimental data to show the community a broader application field of your injection set-up. The field of CTC is of course very interesting but only a limited number of groups is specialized on this in particular whereas generally restricted availability of patient material is a frequent issue. Future will show if systems will be adapted in other labs in your reference. Still I think it is important to ponder about and to try to tackle the problem of injections of such samples, this is why I accept this revised form of the manuscript.

Reviewer 5 Report

Comments and Suggestions for Authors

The authors have answered all of my questions.

  Comments on the Quality of English Language

None.